# AlphaDou: High-Performance End-to-End Doudizhu AI Integrating Bidding

## Abstract

Artificial intelligence for card games has long been a popular topic in AI research. In recent years, complex card games like Mahjong and Texas Hold'em have been solved, with corresponding AI programs reaching the level of human experts. However, the game of Doudizhu presents significant challenges due to its vast state/action space and unique characteristics involving reasoning about competition and cooperation, making the game extremely difficult to solve.The RL model Douzero, trained using the Deep Monte Carlo algorithm framework, has shown excellent performance in Doudizhu. However, there are differences between its simplified game environment and the actual Doudizhu environment, and its performance is still a considerable distance from that of human experts. This paper modifies the Deep Monte Carlo algorithm framework by using reinforcement learning to obtain a neural network that simultaneously estimates win rates and expectations. The action space is pruned using expectations, and strategies are generated based on win rates. The modified algorithm enables the AI to perform the full range of tasks in the Doudizhu game, including bidding and cardplay. The model was trained in a actual Doudizhu environment and achieved state-of-the-art performance among publicly available models. We hope that this new framework will provide valuable insights for AI development in other bidding-based games.

## 1 Introduction

Games can be broadly classified into two categories: perfect-information games (PIGs) and imperfect-information games (IIGs). In PIGs, players can observe all game states, such as in Shogi, Go, and Chess. In contrast, IIGs involve scenarios where participants cannot access complete information about other players, such as in heads-up Texas Hold'em. Reinforcement learning (RL) has been successfully applied to create numerous game AIs. RL algorithms have achieved remarkable success in both PIGs and IIGs, exemplified by AlphaGo (Silver et al., 2016) and AlphaZero (Silver et al., 2017) in Go, AlphaStar (Vinyals et al., 2019) in StarCraft II, OpenAI Five (OpenAI et al., 2019) in Dota 2, Suphx (Li et al., 2020) in Mahjong, Douzero (Zha et al., 2021) in Doudizhu, NukkiAI (Bouzy et al., 2020) in Contract Bridge, and AlphaHoldem (Zhao et al., 2022a) in Hold'em.

However, AI has not performed perfectly in certain gambling games that require bidding. NukkiAI only outperformed professional human players in non-bidding 1v1 Bridge, and Douzero did not consider the bidding phase during training. These AIs function more as playing machines rather than proficient gamblers. The bidding phase contains rich strategic information that significantly influences player strategies. When an opponent has a strong hand, they tend to bid high for higher potential rewards, while players should adopt a conservative strategy to minimize losses. Conversely, when opponents bid low, players can employ more aggressive strategies to increase their gains.

This work aims to develop a high-performance end-to-end Doudizhu AI model that incorporates bidding. Doudizhu, also known as Fighting the Landlord, is the most popular card game in China. Doudizhu is a 3-player IIG where players bid based on their hands, and the winning bidder becomes the Landlord. The remaining players form the Peasants team to oppose the Landlord. If any Peasant player wins, the entire team wins. The Landlord wins double rewards, while each Peasant player receives a single reward if the team wins, and vice versa. Rewards are related to the bid score and the occurrence of "bombs" (four cards of the same rank) or "rockets" during the game. Players with good hands tend to bid high to become the Landlord for higher returns. Moreover, Doudizhu has a

large, flexible, and diverse action space with thousands of possible states ($10^{83}$) and actions (27,472) due to card combinations and complex rules (Zha et al., 2019). Additionally, rewards in Doudizhu are sparse and highly variable, only awarded at the end of the game and influenced by the bidding phase, the number of "bombs" during the game, and "spring" rewards. These characteristics make training a Doudizhu AI extremely challenging, and existing Doudizhu AIs exhibit certain issues.

Previous research on Doudizhu AI has primarily focused on the playing phase, neglecting the bidding phase or employing completely random bid strategies. DeltaDou (Jiang et al., 2019) is the first AI program to achieve human-level performance compared to top human players, using an AlphaZero-like algorithm with Bayesian methods to infer hidden information and sample other players' actions based on their policy networks. However, the vast action space in Doudizhu limits DeltaDou's effectiveness. Douzero introduced Deep Monte Carlo (DMC), which combines the conventional Monte Carlo method with deep neural networks. In a Monte Carlo self-play framework, deep neural networks first estimate the value of each action (Q-value), then select the action with the highest Q-value as a training label or final move. DMC addresses the challenge of Doudizhu's large action space, making training more stable. Doudizhu with DMC successfully outperformed other RL algorithms, including Deep-Q-Learning (DQN) (Mnih et al., 2015; You et al., 2019), Combination Q-Network (CQN) (You et al., 2019), and A3C (Mnih et al., 2016; You et al., 2019). Subsequently, (Wang et al., 2023) noted that the score distribution in gambling games is a combination of winning and losing score distributions, with risk-averse strategies resulting in a significant gap between them. Previous value-based methods directly predicted this combined distribution, leading to high variance and unstable training. They proposed WagerWin (Wang et al., 2023), which introduces probability and value factorization, enabling individual updates of the winning probability, losing Q-value, and winning Q-value. This method stabilizes the training of gambling game AIs. However, WagerWin primarily accelerated AI convergence without significantly improving the optimal policy and winning rate. In addition, several variants of Douzero have been proposed, such as Douzero+ (Zhao et al., 2022b) and Full Douzero+ (Zhao et al., 2024), which incorporate predictions of the opponents' hands based on the original Douzero model. However, these variants do not provide quantifiable improvements over the original Douzero. Mdou (Luo et al., 2024), on the other hand, consolidates the three models of Douzero into a single model for training, resulting in faster convergence. Despite this, the overall performance of the model does not show significant improvement compared to Douzero. RARSMSDou (Luo & Tan, 2024) utilized the PPO framework (Schulman et al., 2017) to enhance Doudizhu AI, addressing the large action space by abstracting actions into several major categories, training a PPO model to select categories, and then training a DMC model to choose actions within the selected category. RARSMSDou outperformed Douzero.

In this paper, we introduce AlphaDou, an end-to-end DouDiZhu AI system that integrates bidding. Our model eliminates the need for abstract state/action spaces or any human-crafted knowledge. It simultaneously estimates both the win rate and the expected value of a given state, enabling it to prune alternative moves based on expectations and select the optimal move strategy based on win rates. Moreover, the model is capable of perceiving bidding outcomes and dynamically adjusting its move strategy accordingly. Extensive experiments demonstrate that our bidding strategy surpasses the performance of bidding networks trained via supervised learning. Additionally, during the cardplay phase, our model consistently outperforms existing DouDiZhu AI systems, whether or not a bidding strategy is employed. The training code for AlphaDou is available.

## 2 THE GAME OF DOUDIZHU

Doudizhu is a three-player card game that is extremely popular in China and is considered a typical gambling game. Among the three players, two are Peasants who need to cooperate to compete against the third player, the Landlord. The game comprises two phases: 1) Bidding and 2) Cardplay.

### 2.1 BIDDING

The Bidding Phase determines the roles of the players. At the start of the game, each player receives seventeen cards from a shuffled deck in a counterclockwise manner, with three cards left in the middle of the table. In the Bidding Phase, a randomly selected player begins the bidding process, followed by the others in sequence. Each player can only bid once, with options to bid 1 point, 2 points, 3 points, or pass. A subsequent player must either choose a higher bid or pass. The first

player to bid 3 points becomes the Landlord, or if all players complete their bids, the player with the highest bid becomes the Landlord, while the other two players become Peasants. The Landlord has the privilege to reveal the three remaining cards for all players to see and then incorporates these cards into his/her hand. Notably, if all three players choose to pass, the game results in a draw, and a new game starts with a fresh deal. The bid score impacts the final game rewards: if the Landlord wins, he/she gains points equal to twice the bid score from both Peasants. Conversely, if any Peasant wins, both Peasants receive points equal to the bid score from the Landlord, who loses double the points. This scenario assumes the absence of Bombs, Rockets, and Spring (refer to the following section).

## 2.2 Cardplay

During the Cardplay phase, players take turns playing cards. Each game consists of multiple rounds, starting with a player playing a valid card combination (e.g., solo, pair). The first round is initiated by the Landlord. Subsequent players must either pass or defeat the previous hand by playing a higher-ranked combination (an action has a rank, refer to Appendix A). The round continues until two consecutive players pass. Then, the player who played the last hand starts the next round. The objective is to clear all cards from one's hand to win. Each "Bomb" and "Rocket" can double the game's stakes. If the Landlord wins, they receive double the rewards, whereas if the Peasant team wins, each Peasant player receives single rewards. Rewards are influenced by the bid score and the presence of Bombs or Rockets. Bombs surpass any action. The only way to defeat a Bomb is with a higher-ranked Bomb or a Rocket. The Rocket is the highest action in the game and can beat any Bomb or action. When a Bomb or Rocket is played, the points at stake double. For instance, if the winning bid is 3 points at the start, it becomes 6 points if a Bomb is played and 12 points if another Bomb is played. With two Bombs played, the Landlord stands to win/lose 24 points, and each Peasant stands to win/lose 12 points. The game concludes when a player clears all their cards. To encourage more aggressive play, Doudizhu includes a "Spring" reward: if throughout the game, the Peasant team makes no plays other than passing, or the Landlord only passes once, it is termed as Spring or Anti-Spring, respectively, doubling the reward (equivalent to playing an additional Bomb).

For more information, readers may also refer to the Wikipedia page on Doudizhu.

## 3 AlphaDou

The goal of AlphaDou is to incorporate the Bid Phase in the training and testing stages of the Doudizhu AI, enabling the AI to fully engage in a complete gambling game. "End-to-end" here means that this framework directly accepts game state information and outputs actions, without requiring handcrafted feature encoding as input or iterative reasoning during decision-making. AlphaDou uses a reinforcement learning (RL) framework to achieve this goal, driven solely by game rewards. The Bid Phase introduces significant variance in game rewards, so we implemented a series of measures to reduce reward variance and make the model's strategy more flexible.

### 3.1 Card Representation and Neural Architecture

For any card combination, excluding jokers, we encode the remaining card combination into a one-hot 4×13 matrix, with 13 columns representing the cards 3, 4, 5, 6, 7, 8, 9, T, J, Q, K, A, 2. The $i$-th row (where $i \in \{0, 1, 2, 3, 4\}$) indicates whether the number of that card type is greater than $i$; if true, it is 1, otherwise it is 0. This is then flattened into a $1 \times 52$ vector, with an additional $1 \times 2$ matrix indicating the presence of the Black and Red Jokers. Figure 1(a) demonstrates this encoding process.

During the bidding phase, we record the observed data as shown in Table 1 and generate a $5 \times 54$ observation matrix for each possible move. All observation matrices are combined into a batch $\times 5 \times 54$ matrix as input data, where the batch size is the number of valid moves.

In the card-playing phase, our recorded data is divided into parts a and b. The observation data in Table 2 is used to generate a $72 \times 54$ observation matrix for each possible move. All observation matrices are combined into a batch $\times 5 \times 54$ matrix as input data part a, where the batch size is the number of valid moves. We also encode the number of bombs played in the game as a one-hot 1

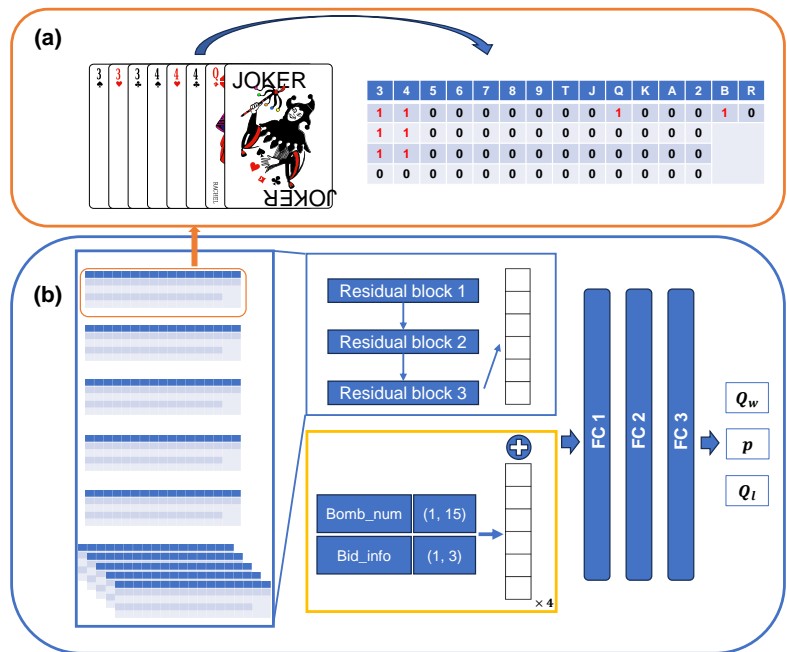

Figure 1: Encoding process of card combinations.

| Observation | Shape | Description |
|---|---|---|
| actions | (1, 54) | Actions from {0, 1, 2, 3} repeated 54 times |
| my_handcards | (1, 54) | Player's current hand |
| 1st bid* | (1, 54) | The bid called by the 1st player, repeated 54 times |
| 2nd bid* | (1, 54) | The bid called by the 2nd player, repeated 54 times |
| 3rd bid* | (1, 54) | The bid called by the 3rd player, repeated 54 times |

* if the score has not been called yet, repeat "-1" 54 times

Table 1: Observation data during the bidding phase.

× 15 matrix, indicating the number of bombs from 0 to 14 that have been played. A 1 × 3 matrix records the bid scores of the first, second, and third players (with -1 if they did not participate in the bidding). The bid data and the bomb count are concatenated and repeated batch times to form a batch × 18 matrix as data part b. Although there might be duplicate inputs in the bid data, the information in data part b directly affects the final game score and is thus included separately as input.

We use six neural networks to model the six positions: "first", "second", "third", "landlord", "landlord_down", and "landlord_up". The "first", "second", and "third" models form the Bid Model, representing the first, second, and third players in the bidding process, respectively. The "landlord", "landlord_down", and "landlord_up" models form the Card Model, representing the Landlord, the player to the left of the Landlord, and the player to the right of the Landlord during the card-playing phase, respectively. The Bid Model and the Card Model have similar network structures. The three neural networks used for bidding in the Bid Model share the same structure, and the three neural networks used for playing cards in the Card Model share the same structure. Figure 1(b) illustrates the neural network structure. To ensure the network gives more importance to inputs that have a

| Observation | Shape | Description |
|---|---|---|
| action | (1, 54) | Actions to be computed by the neural network |
| num_cards_left | (1, 54) | The three one-hot vectors spliced together represent the number of cards remaining with the landlord (1, 20), landlord_up (1, 17), and landlord_down (1, 17) |
| my_handcards | (1, 54) | Player's current hand |
| other_handcards | (1, 54) | The sum of the remaining two players' current hands |
| three_landlord_cards | (1, 54) | Landlord cards not yet played |
| landlord_played_cards | (1, 54) | Cards played by the landlord |
| landlord_up_played_cards | (1, 54) | Cards played by landlord_up |
| landlord_down_played_cards | (1, 54) | Cards played by landlord_down |
| 1st_bid | (1, 54) | The score called by the first player, divided by 3, repeated 54 times |
| 2nd_bid | (1, 54) | The score called by the second player, divided by 3, repeated 54 times |
| 3rd_bid | (1, 54) | The score called by the third player, divided by 3, repeated 54 times |
| spring | (1, 54) | Whether spring bonuses can still be earned |
| card_play_action_seq | (60, 54) | History of cards played (contains 60 historical actions) |

Table 2: Observation data during the card-playing phase.

greater impact on the final score, we repeat the input data part b four times and concatenate it with the residual part of the input. The bidding network input is not divided into parts a and b, so the yellow box region is not present in its structure. The rest of the structure is similar. The parameters of each layer of the neural network are shown in the Table 3.

| Layer | CardModel | | | BidModel | | |
|---|---|---|---|---|---|---|
| | Input | Out | #blocks | Input | Out | #blocks |
| Residual block 1 | $72{\times}54$ | $72{\times}27$ | 3 | $5{\times}54$ | $5{\times}27$ | 3 |
| Residual block 2 | $72{\times}27$ | $144{\times}14$ | 3 | $5{\times}27$ | $10{\times}14$ | 3 |
| Residual block 3 | $144{\times}14$ | $288{\times}7$ | 3 | $10{\times}14$ | $20{\times}7$ | 3 |
| FC 1 | 2088 | 2048 | / | 140 | 256 | / |
| FC 2 | 2048 | 512 | / | 256 | 256 | / |
| FC 3 | 512 | 128 | / | 256 | 128 | / |
| Out layer | 128 | 3 | / | 128 | 3 | / |

Table 3: Parameters of the neural network.

## 3.2 DEEP MONTE-CARLO

Monte Carlo (MC) methods are a class of methods that estimate strategies and value functions by modeling sample paths. Monte Carlo methods are very effective in episodic tasks to estimate the value function by taking every-visit MC approach (Sutton, 1998).

1. **Generating sample trajectories using a specified policy** $\pi$: Starting from an initial state, simulate using the current policy until a terminal state is reached, generating a complete state-action-reward sequence.

2. **Calculating returns and updating** $Q(s,a)$ **values**: For each state-action pair $(s,a)$ in every trajectory, calculate the cumulative return and add it to the return list for that state-action pair. Use the average of these returns to update the $Q(s,a)$ value.

**3. Policy improvement**: For each state, update the policy to select the action with the highest Q value in that state.

$$\pi(s) \leftarrow \arg\max_a Q(s, a)$$

The DMC method has been demonstrated in Douzero to achieve superior results in Doudizhu.

### 3.3 DMC WITH PROBABILITY AND VALUE FACTORIZATION

Considering the significant gap between the distribution of winning scores and losing scores, we perform Value Factorization on the Q-value (Wang et al., 2023). Given that there are no ties in Doudizhu and the outcomes of the game (win or lose) are mutually exclusive, we have:

$$Q(s, a) = p_w(s, a)Q_w(s, a) + (1 - p_w(s, a))Q_l(s, a)$$

where $p_w(s, a)$ is the winning probability given (s,a), and $Q_w(s, a)$ and $Q_l(s, a)$ are the Q-values for winning and losing, respectively.

When updating the Q-Net, we do not directly minimize the Mean Square Error, MSE(reward, predicted Q-value), to update the Q-Net. Instead, we simultaneously optimize the winning probability $p_w(s, a)$, and the Q-values $Q_w(s, a)$ and $Q_l(s, a)$ for winning and losing.

We divide the training data $D$ into two mutually exclusive datasets for winning and losing. For outcome $u$:

$$D = D_w \cup D_l$$
$$D_w = \{(s, a, R_w, u) \mid u = 1\}$$
$$D_l = \{(s, a, R_l, u) \mid u = -1\}$$

$Q_w(s, a)$ is trained using $D_w$, while $Q_l(s, a)$ is trained using $D_l$. The winning probability $p_w(s, a)$ is trained using $\{u\}$ in $D$.

The final loss function is:

$$L = \alpha_1 L_p + \alpha_2 L_q$$

where $\alpha_1$ and $\alpha_2$ are two hyperparameters controlling the weights. The winning probability $p_w(s, a)$ is derived from the neural network output $p(s, a)$:

$$p_w(s, a) = \frac{p(s, a) + 1}{2}$$

The loss function for the probability is:

$$L_p = MSE(p(s, a), u)$$

The loss function for the Q-value is:

$$L_q = \frac{|D_w|}{|D|} MSE_{D_w}(Q_w(s, a), R_w) +$$
$$\frac{|D_l|}{|D|} MSE_{D_l}(Q_l(s, a), R_l)$$

When generating a strategy, we calculate $Q(s, a) = p_w(s, a)Q_w(s, a) + (1 - p_w(s, a))Q_l(s, a)$, and consider whether factors affecting the final reward still exist: whether the spring bonus can no longer be obtained, and whether there are no bomb cards left in this game. If these factors are excluded, theoretically $Q_w(s, a) = |Q_l(s, a)|$, but the absolute values of the neural network outputs are not

always equal, which introduces errors in calculating $Q(s, a)$. In this case, we directly choose the move with the highest winning probability $p_w(s, a)$.

If factors affecting the final reward still exist, we prune the moves based on $Q(s, a)$. We consider moves whose difference from $\max Q(s, a)$ is within a certain percentage range $\rho = 0.05$ as selectable moves, forming the pruned set of selectable moves:

$$A_{cut} \in \{a \mid \left| \frac{Q(s, a) - \max Q(s, a)}{\max Q(s, a)} \right| < \rho\}$$

Then, we choose the move with the highest winning probability $p_w(s, a)$ within $A_{cut}$:

$$a_{best} = \max p_w(s, a), a \in A_{cut}$$

We use the epsilon-greedy method to introduce exploration into the strategy $\pi(s)$ used to generate data. In appendix B, we utilized the win rate model douzero-wp and the expected value model douzero-adp provided by DouZero to verify that our proposed strategy generation method outperforms the "choosing the move with the highest expectation" strategy.

## 4 EXPERIMENTS

In this chapter, we compare the performance of the AlphaDou card-playing model (CardModel) with Douzero and Douzero Resnet. Douzero Resnet is a Doudizhu AI based on the Douzero algorithm, replacing the LSTM neural network in Douzero with ResNet, significantly improving performance compared to Douzero. The weights and code for Douzero Resnet are open-sourced at `https://github.com/Vincentzyx/Douzero_Resnet`. We also compare the AlphaDou bid model (Bid Model) with a supervised learning bid model (Douzero Resnet Bid). The Douzero Resnet Bid we used is derived from Douzero Resnet: fixing the landlord player's hand, randomly distributing 1000 sets of farmer hands and landlord hands, loading the Douzero Resnet model for games, obtaining a mean score from the results of 1000 games, using the hand as input, and the mean score as the label for supervised learning. When applying the model, a threshold is set for bidding: a model output greater than -0.1 bids 1 point, greater than 0 bids 2 points, and greater than 0.1 bids 3 points. The Douzero demonstration website `https://www.douzero.org/bid` also has a bidding model, but its bidding method is not based on a 3-point system, and it does not provide models for tests. Our AI system is trained on a server with 4 Intel(R) Xeon(R) Gold 6330 CPUs @ 2.10GHz and a GeForce RTX 4090 GPU in the Ubuntu 20.04 operating system.

### 4.1 COMPARE BID MODEL TO DOUZERO RESNET BID

Doudizhu has three players, and we categorize them into three positions—first, second, and third—according to the order of bidding. To evaluate the performance of the Bid model, we initially set all three positions to a combination of Douzero and Douzero Resnet Bid, recording the scores for each position after 4000 games (control group). Next, we successively replace the Douzero Resnet Bid at each position with the Bid Model and conduct the same 4000 games to observe whether the scores at each position improve compared to the control group. Since Douzero's card playing is unaffected by the Bid model, we use Douzero as the card-playing model in this experiment.

**Metrics.** Following (Jiang et al., 2019), given an algorithm A and an opponent B, we use two metrics to compare the performance of A and B:

- **WP (Winning Percentage):** The number of games won by A divided by the total number of games.
- **ADP1 (Average Difference in Points 1):** The average difference of points scored per game between A and B. The base point is 1. Each bomb will double the score.
- **ADP2 (Average Difference in Points 2):** The average difference of points scored per game between A and B. The base score is 1 to 3 points, determined by the highest bid during the bidding phase. Each bomb will double the score. Spring bonuses will also double the score.

Additionally, we evaluate each position's:

- **LP (Landlord Percentage):** The number of games in which A became the Landlord Player divided by the total number of games.

- **DR (Draw Rate):** The number of draw games divided by the total number of games.

The test results are shown in Table 4.

| | 1st position | | | 2nd position | | | 3rd position | | | Draw |
|---|---|---|---|---|---|---|---|---|---|---|
| | WP | ADP2 | LP | WP | ADP2 | LP | WP | ADP2 | LP | DR |
| Control | 0.361 | -0.119 | 0.324 | 0.389 | 0.184 | 0.299 | 0.369 | -0.065 | 0.255 | 0.123 |
| 1st | **0.411** | **0.014** | **0.423** | 0.420 | 0.115 | 0.285 | 0.401 | -0.129 | 0.243 | **0.049** |
| 2nd | 0.387 | -0.116 | 0.321 | **0.416** | **0.266** | **0.377** | 0.386 | -0.150 | 0.224 | **0.078** |
| 3rd | 0.387 | -0.207 | 0.315 | 0.415 | 0.099 | 0.290 | **0.415** | **0.108** | **0.342** | **0.054** |
| 1st & 2nd | **0.424** | **0.014** | **0.416** | **0.435** | **0.193** | **0.331** | 0.404 | -0.207 | 0.223 | **0.030** |
| 1st & 3rd | **0.418** | **-0.048** | **0.387** | 0.423 | 0.031 | 0.281 | **0.424** | **0.018** | **0.311** | **0.021** |
| 2nd & 3rd | 0.400 | -0.198 | 0.314 | **0.432** | **0.191** | **0.360** | **0.419** | **0.007** | **0.294** | **0.032** |
| all | 0.426 | -0.035 | 0.384 | 0.436 | 0.127 | 0.324 | 0.421 | -0.090 | 0.282 | **0.010** |

Table 4: Performance of Bid Model against Douzero Resnet Bid by playing 4,000 randomly sampled decks. The Control group means use Douzero Resnet Bid models in all the 3 positions. For each experimental group, we changed some of the positions to the Bid Model. Results where the experimental group outperforms the control group are highlighted in boldface. The sum of WP is not 1 because two players win when the Peasants Team wins.

For each test group, the Bid Model shows significant improvement in WP, ADP2, and LP compared to Douzero Resnet Bid, while also achieving a lower Draw Rate. In Doudizhu, the Landlord Player wins double the rewards, so accurately determining whether a player should become the Landlord Player is crucial for scoring. The Bid Model is more aggressive, tending to become the Landlord Player more often, whereas Douzero Resnet Bid is more conservative. One reason is that the Bid Model adjusts its bids by considering the bids of other players; when opponents bid low, the Bid Model may bid high even if the player's hand is not exceptionally good but relatively better than the opponents' hands.

When two positions are replaced with Bid Models, the ADP and LP of the position still using Douzero Resnet Bid significantly decrease, indicating that the more accurate judgment of the Bid Models exploits the Douzero Resnet Bid.

In the Appendix C, we provide a detailed analysis of the performance of the Bid Model in real gameplay scenarios, illustrating the strategic improvements it offers compared to Douzero Resnet Bid. The Bid Model demonstrates the ability to adjust its strategy based on the opponent's actions and may also adopt a more aggressive bidding approach to force the opponent into retreat (bluffing).

## 4.2 COMPARE CARD MODEL TO BENCHMARKS WITH RANDOM BIDDING PHASE

To evaluate the performance of the Card Model, we followed the approach of (Jiang et al., 2019) and Douzero (Zha et al., 2021), initiating a competition between the Landlord and the Peasants. We reduce variance by playing each deck twice. Specifically, for two competing algorithms A and B, they will first play with A as the Landlord and B as the Peasants for a given deck. Then, they swap roles, with A as the Peasants and B as the Landlord, and play the same deck again. A total of 4,000 games were conducted. Considering that the Bid result is random, we set the initial score of the game to 2 points for the Landlord's win and 1 point for the Peasants' win, with each bomb doubling the final score (we define this scoring method as ADP1). The Card Model needs to decide the playing strategy based on the bidding process, and the random bidding process will lead to a decline in model performance because the random testing deck distribution deviates from the training process.

Table 5 shows the results. As of the completion of this paper, RARSMSDou is the strongest publicly available Doudizhu model. In a 1000-game test with Douzero using random bidding, it achieved a win rate of 0.582 and an ADP1 of 0.414. Although we did not directly test AlphaDou against

RARSMSDou, both were tested against the baseline Douzero. AlphaDou performed better than RARSMSDou in the random bidding test.

| A \ B | Card Model | | Douzero Resnet | | Douzero | |
|---|---|---|---|---|---|---|
| | WP | ADP1 | WP | ADP1 | WP | ADP1 |
| Card Model | - | - | **0.522** | **0.103** | **0.597** | **0.434** |
| Douzero Resnet | -0.478 | -0.103 | - | - | **0.570** | **0.269** |
| Douzero | 0.403 | -0.434 | 0.423 | -0.269 | - | - |

Table 5: Performance of Card Model against Douzero Resnet and Douzero by playing 10,000 randomly sampled decks with random bidding phase. Algorithm A outperforms B if WP is larger than 0.5 or ADP is larger than 0 (highlighted in boldface).

### 4.3 COMPARE CARD MODEL TO BENCHMARKS WITH BIDDING PHASE

We conducted another 4,000 matches and with the bid model set to Bid Models. The game scores are divided into ADP1 and ADP2. ADP1 is consistent with the one mentioned above, and ADP2 is calculated based on the results of the Bid Models. For example, if the landlord wins with 3 points, the landlord scores 6 points, the peasants score 3 points, and each bomb will double the final score. The results are shown in the table 6.

| | Card Model | | | Douzero Resnet | | | Douzero | | |
|---|---|---|---|---|---|---|---|---|---|
| | WP | ADP1 | ADP2 | WP | ADP1 | ADP2 | WP | ADP1 | ADP2 |
| Card Model | - | - | - | **0.544** | **0.315** | **0.738** | **0.620** | **0.576** | **1.585** |
| Douzero Resnet | 0.456 | -0.315 | -0.738 | - | - | - | **0.581** | **0.314** | **0.937** |
| Douzero | 0.383 | -0.576 | -1.585 | 0.419 | -0.314 | -0.937 | - | - | - |

Table 6: Performance of Card Model against Douzero Resnet and Douzero by playing 4,000 randomly sampled decks with bidding phase. Algorithm A outperforms B if WP is larger than 0.5 or ADP is larger than 0 (highlighted in boldface).

The Card Model still dominates all other algorithms. Compared to the random Bidding Phase, the WP of the Card Model against Douzero Resnet and Douzero has significantly improved. This indicates that the Card Model can adjust its playing strategy based on the Bid results to achieve higher returns. Notably, with the Bidding Phase, the WP of Douzero Resnet against Douzero also increased (0.5809 > 0.5702), but Douzero Resnet does not adjust its playing strategy based on the bid results. We believe this is because the bid results provided by the Bid Model are favorable to the landlord, and if a model's landlord strength is very strong, its overall win rate will correspondingly increase.

Table 7 shows the WP and ADP of the Card Model (Landlord) and Douzero Resnet (Landlord) against Douzero (Peasant). It can be seen that the strength of Douzero Resnet (Landlord) is quite similar to that of the Card Model (Landlord). Therefore, after adjusting the bid strategy, the overall win rate of Douzero Resnet against Douzero will increase. Correspondingly, we find that although the strength of Douzero Resnet (Landlord) is similar to that of the Card Model (Landlord), the strength of Douzero Resnet (Peasant) is much weaker than that of the Card Model (Peasant). This may be because the Landlord side only needs to consider confrontation, while the Peasant side needs to consider cooperation, making it easier for the Landlord model to converge.

| | Random | | Bidding | |
|---|---|---|---|---|
| | WP | ADP1 | WP | ADP2 |
| Card Model | 0.514 | -0.048 | 0.785 | 4.645 |
| Douzero Resnet | 0.490 | -0.209 | 0.771 | 4.296 |

Table 7: Performance of Card Model (Landlord) and Douzero Resnet (Landlord) against Douzero (Peasant) by playing 4,000 randomly sampled decks.

In the Appendix D, we provide a detailed analysis of the performance of the Card Model in real gameplay scenarios. Compared to Douzero and Douzero Resnet, the Card Model is more adept at accurately assessing the current state, making it better at predicting the opponent's hand and discerning their intentions.

## 5 CONCLUSION

The game of Doudizhu is an extremely challenging incomplete information game. It has a vast state/action space and unique characteristics involving reasoning about competition and cooperation, making the game particularly difficult to solve. Research on Doudizhu typically simplifies the game by not considering the bidding phase and the "spring" bonus, as including these factors increases the variance in rewards, making the model harder to converge. Additionally, the inclusion of bidding can cause deviations in the card distribution compared to when bidding is not included.

This paper first incorporates factors like bidding and the spring bonus to make the research environment more closely resemble the actual Doudizhu game environment. Secondly, it modifies the Deep Monte Carlo algorithm framework, using reinforcement learning to obtain a neural network that simultaneously estimates win rates and expectations. The action space is pruned using expectations, and strategies are generated based on win rates. This modification allows the DMC algorithm to produce strategies that are not solely dependent on value (expectation) but also consider win rates, resulting in a state-of-the-art (SOTA) Doudizhu reinforcement learning model, which we named AlphaDou. We compared AlphaDou with the baseline program DouZero, achieving a win rate of 0.6167 in an environment with bidding. Even when there are differences between the training environment with bidding and the testing environment without bidding, AlphaDou still achieved a win rate of 0.5970 and an average score per game of 0.4343, making it the SOTA RL model. RL models trained in complex environments can also perform excellently in more simplified environments.

The framework of AlphaDou may offer valuable insights for other activities that require balancing event success rates and expected returns, such as bridge games or bidding strategies in advertising. Additionally, the decomposition of values presents a potential advantage: it could allow for a better understanding of AI model behavior (whether the model leans more towards win rates or returns). A more detailed decomposition of values could significantly enhance the interpretability of the model's decisions, and there is a possibility that, by integrating large language models, the AI could explain its decision-making process by itself, even though this might not necessarily improve the model's performance.

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

## A  THE COMBINATIONS AND RANKS OF CARDS

One of the challenges in the game of Doudizhu is the vast state/action space, which includes numerous card combinations. For certain categories, players can choose a "kick-out" card, which can be any card from their hand, directly leading to a large action space. For the landlord player, the winning condition is to play all their cards, while farmer players do not always need to play all their cards; their teammate clearing their hand also signifies victory. This requires considering using larger cards as kick-out cards and retaining smaller cards to coordinate with the teammate's plays. Players need to carefully strategize their moves to win the game. The classification of card types in Doudizhu is shown in the Table 8. Note that "Bombs" and "Rockets" break category rules and can dominate all other categories.

## B  MIXTURE OF THE POLICY MAKES THE AI STRONGER

Douzero has open-sourced two model weights: douzero-wp, which uses win rate as the reward, and douzero-adp, which uses expectation as the reward. Douzero generates strategies based on the maximum output of douzero-adp. We propose the following two methods to consider both douzero-wp and douzero-adp models simultaneously to generate strategies:

1. **Bomb check**: Check for factors that influence the final reward. If none exist, choose the move with the highest win rate based on douzero-wp output. Factors influencing the final reward include spring reward and bomb reward. Douzero does not consider the spring reward, so this step is to determine the presence of a bomb.

2. **Mixed strategy (Mix)**: Prune moves based on the expectation derived from douzero-adp. We consider moves with an expectation difference within a certain percentage range ($\rho = 0.05$) from the maximum expectation as viable moves. Then, select the move with the highest win rate among the viable moves.

We can derive four different RL models for generating strategies: Douzero, Douzero with only Bomb check (Bomb check), Douzero with only mixed strategy (Mix), and Douzero with Bomb check followed by mixed strategy (Bomb check & Mix). We tested the performance of these four models against Douzero in fixed 4000 game scenarios at different positions (landlord, farmer). The specific results are displayed in the Table 9. It can be seen that both Bomb Check and Mixed strategy yield better strategies than the standalone douzero-adp. Bomb Check followed by Mixed strategy achieves the best strategy.

## C  CASE STUDY: BID MODEL VS DOUZERO RESNET BID

In these cases, we use the following abbreviations: "P" for "Pass", "T" for card "10", "J" for Jack, "Q" for Queen, "K" for King, "A" for Ace, "B" for Black Joker, and "R" for Red Joker. Each action is represented as "position: action," where "position" can be "L" for Landlord, "D" for Peasant-Down, or "U" for Peasant-Up. For example, "L:TT" denotes the Landlord playing a Pair (10 10), and "D: 22" indicates Peasant-Down playing a Pair (22). The actions are separated by commas (e.g., "L:J,D:Q,U:Pass").

Compared to threshold bidding, reinforcement learning bidding has a more flexible handling of different bidding situations. Here, we analyze a hand with the cards 333444569TTJJQKK2. The hand score given by Douzero Resnet Bid is -0.987, indicating that this hand is very weak. Firstly, the only high card is 2, and the absence of 7, 8, A, B, and R means that there is a high probability that other players' hands form bombs. Using Douzero Resnet Bid, the choice would be "0 points". For the same hand, the Bid Model gives different bidding scores based on the bidding order. In different situations, the Bid Model's bidding strategy varies, as shown in Table 10. The Bid Model tends to bid 2 points in the first position, 0 points in the second position, and 3 points in the third position (pass is chosen only if 2 points were bid in the first position).

In first position, the model would bid 2 points, but the model prediction would be more inclined to say "the other player will deal the landlord for 3 points". Bidding 2 points is close to gaining -2.998 points, but choosing a different strategy would result in a greater loss of points. Choosing to bid 2 points also signals to possible teammates that my hand is neater and easier to complete. This is not a

| Category of Actions | Description | Num |
|---|---|---|
| Pass | Not play cards | 1 |
| Solo (F) | Any single card.
3<4<5<6<7<...<K<A<2<B<R | 15 |
| Pair (F) | Two same cards.
33<44<55<66<...<KK<AA<22 | 13 |
| Trio (F) | Three same cards.
333<444<555<...<KKK<AAA<222 | 13 |
| Trio-Solo (F) | A Trio and a Solo.
333? <444?<555?<...<AAA? <222? | 182 |
| Trio-Pair (F) | A Trio and a Pair.
333* <444*<555*<...<AAA* <222* | 156 |
| Bomb (F) | Four same cards.
3333<4444<5555<...<AAAA <2222 | 13 |
| Rocket (F) | Black and Red Jokers | 1 |
| Quad-Solo (F) | Bomb with 2 additional Solos.
3333??<4444??<...AAAA??< <2222?? | 1326 |
| Quad-Pair (F) | Bomb with 2 additional Pairs.
3333**<4444**<...AAAA**< <2222** | 856 |
| Chain-Solo (V) | Least 5 consecutive cards
34567<45678<...<9TJQK<TJQKA
345678<456789<...<89TJQK<9TJQKA | 36 |
| Chain-Pair (V) | Least 3 consecutive cards
334455<445566<...<QQKKAA
33445566<44556677<...<JJQQKKAA | 52 |
| Plane (V) | Least 2 consecutive Trios
333444<444555<...<KKKAAA
333444555<444555666<...<QQQKKKAAA | 45 |
| Plane-Solo (V) | Plane with each Trio has a distinct Solo.
333?444?<444?555?<...<KKK?AAA?
333?444?555?<444?555?666?<...<QQQ?KKK?AAA? | 21822 |
| Plane-Pair (V) | Plane with each Trio has a distinct Pair.
333*444*<444*555*<...<KKK*AAA*
333*444*555*<444*555*666*<...<QQQ*KKK*AAA* | 2939 |

Table 8: Actions and Their Ranks in Doudizhu. Doudizhu uses a 54-card deck, which includes 3, 4, 5, 6, 7, 8, 9, 10 (T), Jack (J), Queen (Q), King (K), Ace (A), 2, Black Joker (B) and Red Joker (R). Suits are irrelevant. "?" and "*" denote any Solo or Pair, respectively. "F" and "V" denote fixed-length action and variable-length action, respectively. This table is cited from (Luo & Tan, 2024).

strong hand, but there is an airplane (333444), which makes the hand look neat and also means that there will be few small cards in the other players' hands. In a state where there are very few small cards and a lot of big cards, the option to bid 3 points allows the player to get maximum value. If the landlord is sold for 2 points, it means that the big cards are in the landlord cards, or that the other player's hand is so untidy that it will take many hands to play it out, In which case, the chances of winning are higher, being able to overcome a weaker hand by utilizing a weaker hand.

In the second position, if the first bidder doesn't show strong card power (less than 2 points) and the strong card power isn't in their own hand, it must be with the third bidder. The model then evaluates

|  | Landlord | | Farmer | | Overall | |
|---|---|---|---|---|---|---|
|  | WP | ADP1 | WP | ADP1 | WP | ADP1 |
| Douzero | 0.434 | -0.391 | 0.566 | 0.391 | 0.5 | 0.0 |
| **Bomb check** | **0.442** | **-0.362** | **0.578** | **0.449** | **0.509** | **0.043** |
| **Mix** | **0.446** | **-0.331** | **0.581** | **0.460** | **0.513** | **0.064** |
| **Bomb check & mix** | **0.452** | **-0.306** | **0.586** | **0.482** | **0.519** | **0.088** |

Table 9: Performance of Different Strategies

the win rate as extremely low ($\approx 0.1$). The model predicts $|Q_l| \approx 6$ and $Q_w \approx 3$, indicating that the third bidder is very likely to have a bomb and will bid 3 points to become the Landlord. With the bomb in the Landlord's hand, even if they sense defeat, they won't use the bomb to avoid doubling the loss. Thus, even winning yields only 3 points. Bidding 3 points to become the Landlord in this situation results in a significant negative return, while bidding 0 points or any other score could lead to a misjudgment by the Peasant teammate about the hand strength. If the first bidder bids 2 points, it indicates some card power, reducing $|Q_l|$ and increasing the win rate. Still, bidding is not advisable as the Landlord will likely lose to the first bidder. This analysis for the second position is based on the rule that the Landlord loses double the points compared to the Peasant. If the loss points for the Landlord and Peasant were the same, the strategy could be to bid 3 points and become the Landlord, especially if the first bidder bids 0 points. This strategy is common in professional JJ Doudizhu tournaments where the Landlord and Peasant lose the same points upon failure.

In the third position, the model tends to bid 3 points, with $Q_w$ approaching 9 points. The model believes that high card power (Rocket) is in the landlord cards, and the remaining players' card types are not neat, making it difficult to handle the airplane card type (333444) and potential consecutive pairs (99TTJJ, TTJJQQKK, etc.). In this case, bidding 3 points to become the Landlord could result in gaining a bomb or rocket along with a Spring, cleverly utilizing the bidding position to allow weak card power to exploit even weaker card power.

| Position | First | Second | Agent | Win rate | $Q_w$ | $Q_l$ | $Q$ |
|---|---|---|---|---|---|---|---|
| First | / | / | 2 | 0.2724 | 4.4712 | -5.7912 | -2.9976 |
| Second | 0 | / | 0 | 0.1122 | 3.7488 | -5.8030 | -4.7328 |
| Second | 1 | / | 0 | 0.1203 | 3.8616 | -6.0600 | -4.8672 |
| Second | 2 | / | 0 | 0.2407 | 3.3096 | -4.3200 | -2.4816 |
| Third | 0 | 0 | 3 | 0.5154 | 9.3168 | -7.7808 | 1.0296 |
| Third | 0 | 1 | 3 | 0.4742 | 9.4392 | -7.8432 | 0.3528 |
| Third | 0 | 2 | 3 | 0.4378 | 8.5080 | -7.5552 | -0.5232 |
| Third | 1 | 0 | 3 | 0.4907 | 9.4536 | -7.6920 | 0.7200 |
| Third | 1 | 2 | 3 | 0.4281 | 7.9008 | -7.1016 | -0.6792 |
| Third | 2 | 0 | 0 | 0.2477 | 2.2656 | -2.7432 | -1.5024 |

Table 10: The Bid Model's strategy for bidding points in different situations

# D   CASE STUDY: CARD MODEL VS DOUZERO RESNET VS DOUZERO

Figure 2 shows the cards held by the landlord, the landlord's next player, and the landlord's previous player. The landlord does not have any joker cards, but their hand is well-structured and strong. The landlord chooses to play 44, which brings the game to the first critical point: the landlord's next player Douzero plays KK, while Card Model and Douzero Resnet choose to play AA. Playing KK allows the landlord to regain card rights with AA, whereas playing AA prevents the landlord from taking card rights.

First, let's analyze the scenario where the farmer plays KK and the landlord regains card rights with AA. In this situation, the landlord has the advantage, with Douzero Resnet and Douzero opting to play 5557, while Card Model chooses to play 888999TJ, as shown in Figure 3. In this scenario, if the opponent plays QQQ5 after 5557, the only way to regain card rights is by playing 222K. The

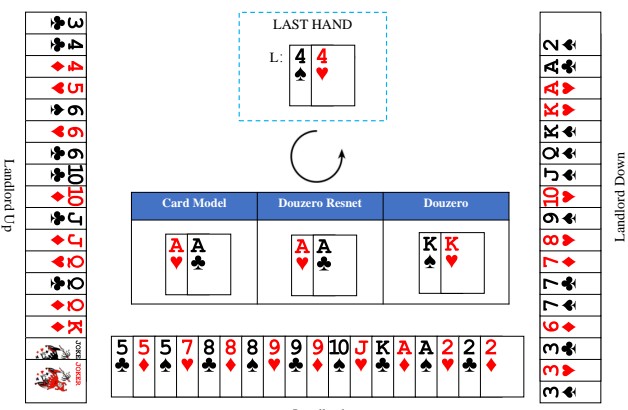

Figure 2: Landlord's, landlord's next player's, and landlord's previous player's hands

landlord's previous player then uses the rocket to obtain forced card rights, leaving the landlord with 7888999K, resulting in failure and bomb penalty for the landlord. Conversely, Card Model's play of 888999TJ, an airplane type, is a rare hand. The opponent has a low probability of suppressing it. If the farmer uses the rocket to gain forced card rights, the landlord's remaining hand is 5557K222. With three 2s being the highest cards, the landlord can still win and receive a bomb reward. If the farmer opts to pass against the airplane type, the landlord can play 5557 and have 222K left. The farmer can only prevent the landlord from playing all their cards in the next hand by using the rocket. However, using the rocket leaves the landlord with three 2s, and the farmer cannot win. The farmer can only avoid using the rocket to evade the bomb penalty. The Card Model landlord values card rights more and plays more conservatively, leading to a higher win rate.

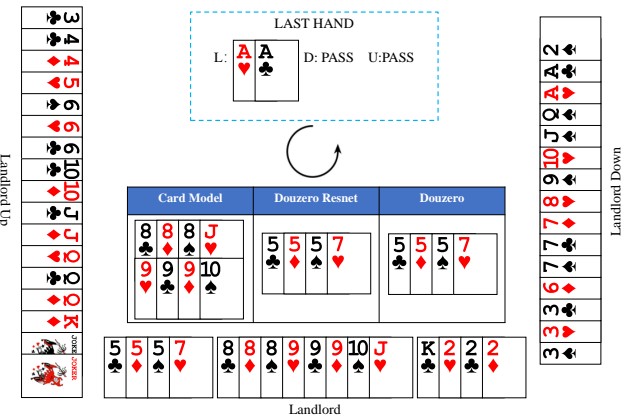

Figure 3: Play scenarios after the landlord regains card rights

From the analysis above, it is clear that the farmer's choice to play KK leads to quick failure. Only by playing AA can the farmer have a chance to win. Card Model and Douzero Resnet are more sensitive to potential dangers. After the landlord's next player plays AA and gains card rights, the game reaches the second critical point, as shown in Figure 4. Douzero Resnet opts to play 89TJQ to gain card rights before playing 3336, while Card Model directly plays 3336.

Analyzing Douzero Resnet's play, playing 89TJQ reduces hand complexity. When 3336 is played next, the landlord plays 555T, leaving the landlord's next player with 777KK2. If they play 7772, leaving KK, the landlord can play 2227. Even if the landlord's previous player uses the rocket for forced card rights, the landlord's remaining AA will be the highest cards and win, earning the bomb reward. Returning to 777KK2, if 777K is played, leaving K and 2, it can secure a win. However, leaving two single cards is unwise, especially when neither K nor 2 is the highest card (the joker

hasn't appeared yet). This incomplete information game makes the 777K strategy unlikely, leading to the farmer's almost certain failure.

Now, consider Card Model's play of directly playing 3336. After the landlord plays 555T, the landlord's next player is inclined to play 777K. This is because the remaining K can combine with 89TJQ to form 89TJQK, retaining the single 2, ensuring the farmer's victory. The Card Model farmer can maintain hand diversity in complex situations, keeping more possibilities open, which also results in a higher win rate.

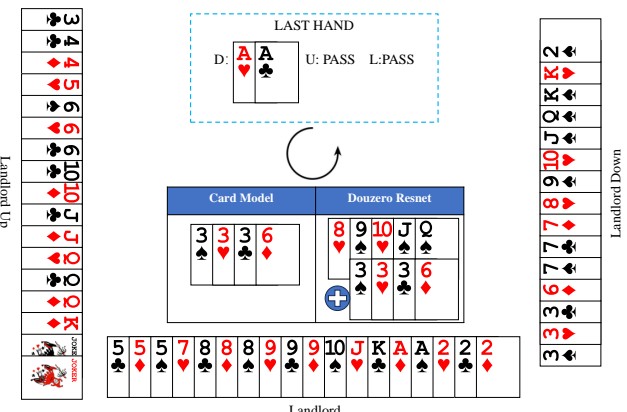

Figure 4: Second critical point in the game

