# OpenReview forum: "AlphaDou: High-Performance End-to-End Doudizhu AI Integrating Bidding"
_ICLR.cc/2025/Conference — ICLR 2025 Conference Withdrawn Submission_

### Official Review · Reviewer_jict · 2024-10-30

**Soundness:** 2
**Presentation:** 2
**Contribution:** 2
**Rating:** 3
**Confidence:** 5

**Summary:**

This paper introduce AlphaDou, an end-to-end DouDiZhu AI system that integrates bidding. This model is capable of perceiving bidding outcomes and dynamically adjusting its move strategy accordingly. Compared to previous algorithms for DouDiZhu, AlphaDou demonstrates enhanced capabilities in the experiments.

**Strengths:**

1. This paper provides a comprehensive summary of the related work on AI for DouDiZhu.
2. This paper is a good AI project implementation in the application of DouDiZhu.

**Weaknesses:**

1. The clarity of the paper's writing is insufficient; please refer to the Question.
2. The experimental validation is not sufficiently robust.
3. The input of the model includes the number of bombs played in the game, and I think this point involves the use of expert knowledge. An algorithm is end-to-end if only the direct observation is used as the input with post-processing.

**Questions:**

1. To what extent does considering bidding affect the complexity of the game of DouDizhu?
2. You should directly competing AlphaDou with RARSMSDou since RARSMSDou is the strongest publicly available Doudizhu model. Different models employ distinct strategies. A higher win rate against the same benchmark by one model does not necessarily imply that it can defeat another model.
3. In line 152, I am not entirely clear on why "i" can equal 4. Since each card has a maximum count of 4, it is impossible for the count of cards to be greater than i=4.
4. In line 389, it seems that the reported results in the column of the 1st position is worse than the results in the column of the 2nd position, is it correct?

---

> ### Author Response · Authors · 2024-11-13
>
> 1.	To what extent does considering bidding affect the complexity of the game of DouDizhu?
>
> Bidding provides additional information that influences the model's decision-making. The bidding process prompts the model to further assess the card types held by the Landlord or teammates. This includes, but is not limited to:
> (a) If the Landlord wins with a low bid, there is a lower probability of bombs in their hand, indicating weaker card strength; conversely, a higher bid suggests stronger cards.
> (b) After becoming the Landlord, if someone else also bid, the model should be more cautious of the risk of bombs.
> (c) As a Peasant, if a teammate placed a bid, the model should consider greater coordination with them.
> Bidding also affects the card distribution differently than in games without bidding. Players with stronger hands are more likely to become the Landlord, and games with a bidding stage tend to have a higher Landlord win rate, whereas Peasants win more often in games without bidding. This also impacts the model’s decision-making. In certain scenarios, AlphaDou may adopt an aggressive playstyle assuming the Peasant’s hand is weaker, similar to professional human players. However, models like Douzero or Douzero Resnet, being more concerned about strong Peasant hands, may miss opportunities as a result.
>
> 2.	You should directly competing AlphaDou with RARSMSDou since RARSMSDou is the strongest publicly available Doudizhu model. Different models employ distinct strategies. A higher win rate against the same benchmark by one model does not necessarily imply that it can defeat another model.
>
> Before AlphaDou, the strongest AI was RARSMSDou; however, since it is not open-sourced, direct comparison with it is challenging. We manually implemented an AI similar to what was described in the paper, but we were unable to reproduce its results. We plan to add results of comparisons with open-source AIs, such as perfectDou or AP-MCTS (Zhang, Yunsheng, et al. "Combining Tree Search and Action Prediction for State-of-the-Art Performance in DouDiZhu." IJCAI. 2021).
>
> 3.	In line 152, "i = 4" is an error, and it does not match the illustration in Figure 1(a). I will correct this mistake in the upcoming version of the submission.
>
> 4.	In line 389, it seems that the reported results in the column for the 1st position are worse than those in the column for the 2nd position. Is this correct?
>
> Yes, this is correct, as this table should be viewed vertically. It uses a controlled variable method, comparing the replacement of one model or two models against the control. Changing the bidding strategy does not necessarily suppress the opponents' win rates. In fact, the addition of the bid model increases the win rates of the remaining two opponents, as the bid model is more willing to take on the role of the Landlord (reflected in a higher Landlord Percentage, or LP) compared to the original strategy. A higher LP tends to lower the Landlord’s win rate, as the Landlord’s loss results in a win for both Peasants. However, through vertical comparison, it’s evident that the bid model consistently achieves a higher score than it did before the strategy change.
>
> 5.	The input of the model includes the number of bombs played in the game, and I think this point involves the use of expert knowledge. An algorithm is end-to-end if only the direct observation is used as the input with post-processing.
>
> I don’t believe that the "number of bombs played in the game" involves expert knowledge, but it does help the model understand the game. The same applies to "remaining cards" and "cards played by each position." For the former, it requires subtracting one’s own cards and the cards others have played from the full deck, while for the latter, it involves accumulating the cards played by each position. I also directly provided these results to the model.

---

### Official Review · Reviewer_kf7g · 2024-10-31

**Soundness:** 2
**Presentation:** 2
**Contribution:** 2
**Rating:** 3
**Confidence:** 4

**Summary:**

This paper introduces AlphaDou, an end-to-end trained reinforcement learning (RL) agent that plays the game of Doudizhu. AlphaDou consists of a bid model and a card model that handles the bidding and playing phases separately. Central to the technique of AlphaDou are Deep Monte Carlo (DMC), value factorization and action pruning. The experiments demonstrate that AlphaDou achieves a higher win rate and average point difference against previous models, especially after incorporating the bid model.

**Strengths:**

- The AlphaDou framework is straightforward and likely easy to implement.
- The bid model is a novel artifact responsible for improving the overall playing strength of AlphaDou.
- AlphaDou appears to be a more powerful agent than the previous DouZero agent in terms of winning rate and point difference.

**Weaknesses:**

- The contribution of AlphaDou requires more explanation. Except for the bid model, it directly combines two ideas already explored -- Deep Monte Carlo and value factorization, leaving the contribution of this work unclear.
- The paper does not have a background section, leaving several notations undefined. The authors should not assume that all readers have the necessary background in RL or understand the notations without proper definitions. In addition, the equation numbers are missing.
- Some claims made by the authors are merely speculations without evidence. For instance, "when opponents bid low, the Bid Model may bid high even if the player’s  hand is not exceptionally good but relatively better than the opponents’ hands."
- The improvement of AlphaDou over previous baselines is marginal and lacks an ablation study.

Some minor issues:
> ... complex card games like Mahjong and Texas Hold’em have been solved

I believe we say a game is solved when we at least discover the game-theoretic outcome when all players play optimally. Under this definition, Mahjong and Texas Hold'em are not solved games.

**Questions:**

- Since $p_w(s, a)$ models the winning probability, the cross entropy loss is a natural choice. Why did you choose the mean square error over the cross entropy loss?
- What is your justification for the action pruning method? Do you have experiments demonstrating its effectiveness?

---

> ### Author Response · Authors · 2024-11-13
>
> •	Regarding the First Question：
>
> Estimating the win rate is not a binary classification problem, so using BCE loss isn’t ideal for reinforcement learning tasks. Most models, like AlphaGo, DouZero use MSE, while supervised tasks like Stockfish use BCE. Since you’ve raised concerns about MSE, I’ll explain briefly why it’s better suited for these tasks.In binary classification, when a model outputs a value like 0.6 or 0.99, it reflects the model’s confidence. However, in reinforcement learning, the win rate is a probability, not a confidence level. If the win rate is 0.6, it means there's a 40% chance of losing, and the model should output 0.6, not 1. Similarly, if the win rate is 0.99, the model should output 0.99, not 1.
> Why BCE Loss is not ideal for reinforcement learning: On the other hand, MSE works better for win rate estimation, as it gives smaller gradients in highly probable outcomes (like when a win or loss is nearly certain) and allows the model to adjust slightly for rare events (like comebacks).Using 1 and -1 as rewards provides better symmetry, which is natural for games with different roles (like in Go, where the output magnitude is similar for both black and white players).
> (a) BCE loss works well in classification tasks because it gives large gradients when there’s a misclassification (like when the label is 0 but the model outputs 1).
> (b) But in reinforcement learning, if the model's output is close to the label (e.g., 0.99 for a 99% win rate), the model shouldn't overreact or try to correct it too forcefully, especially in rare events (like a comeback).
>
> •	Regarding the Second Question:
>
> For the method of "selecting moves based on win rate after expectation pruning" and the "bombs check," we conducted an ablation study, which is included in the appendix B. This is mentioned in line 339 of the main text and has been proven to be effective.
>
> From a theoretical perspective, we combine win rate and expected value, clearly choosing win rate as the basis for decision-making. In zero-sum card games, previous research has rarely (in fact, I haven't seen any) used win rate as a standard for selecting moves. However, the model's estimation of win rate is often more accurate; the labels for win rate are 1 and -1, while labels for scores can be 3, -3, 6, -6, and so on. This means the uncertainty in estimating win rate is smaller than that of estimating scores. To give a simple example: a move with a win rate of 0.99 corresponds to an expected value of 2.98 points, while a move with a win rate of 0.98 corresponds to an expected value of 2.99 points. In a zero-sum game, choosing a move with an expected value of 2.98 instead of 2.99 results in a loss of 0.01 points. Considering the uncertainty, it is clear that the move with a win rate of 0.99 should be chosen because the actual loss is 0.01 + δ (uncertainty). This is the core idea behind the improvement and a key reason for AlphaDou's success. We require the model to use win rate-based decision-making in the absence of "Boom Check" passing, accounting for uncertainty. Incorporating this idea into the data generation process during training helps the model generate higher-quality data for gradient descent.
>
> •	Explanation of the Significance of Experimental Results
>
> The experimental results show that AlphaDou's performance improves significantly; DouZero Resnet achieves a winning percentage of 0.57 in 2021 and is ranked number one in the Botzone for three consecutive years, while RARSMSDou improves this to 0.582 and AlphaDou to 0.596. In the bid test set, the winning percentage reaches 0.620, about 0.4 higher than RARSMSDou's. PerfectDou’s win rate against DouZero is 0.54.
>
> •	Explanation of the Ablation Study Question:
>
> In the Douzero ResNet project, it’s been shown that a convolutional network outperforms Douzero’s LSTM network. Our architecture is slightly different, but it’s based on Douzero ResNet, so the difference is not significant. Providing an exact analysis of the model architecture's impact would be difficult and costly.
> Regarding the Bid phase, introducing it definitely improves performance by providing additional information. Our experiments show that while the model performs better with bidding than with random bidding, it still achieves state-of-the-art results. Random bidding does cause some performance degradation due to bias between the test and training sets. However, whether training without the bidding phase would improve performance on a non-bidding test set remains unclear. That said, this isn’t the focus of our research. The main goal is to develop a state-of-the-art model that incorporates the bidding phase, as it aligns with the actual rules of DouDiZhu.
>
> •	We agree with your understanding of "solved" and will modify. We will also modify the formula section.

---

### Official Review · Reviewer_cwtH · 2024-11-01

**Soundness:** 2
**Presentation:** 2
**Contribution:** 2
**Rating:** 3
**Confidence:** 4

**Summary:**

This paper developes a new Doudizhu AI, which modifies the Deep Monte Carlo algorithm framework (i.e. DouZero) by using reinforcement
learning to obtain a neural network that simultaneously estimates win rates and expectations. Stronger performance is achieved against DouZero.

**Strengths:**

Stronger performance is achieved against DouZero and its variants.

**Weaknesses:**

- Weak experiments. More recent SOTA Doudizhu AIs should be included as the baseline methods.
- The reason why the proposed method AlphaDou is better than DouZero is unclear.
- The improvements (both in terms of methodology and experimental results) of the proposed method AlphaDou over DouZero seem marginal.

Minor:
- Section 2, The game of Doudizhu, is suggested to be moved to the appendix.
- The end of Introduction. The training code for AlphaDou is available. Please attach the code link if you claim it available.
- Missing previous related work on Doudizhu AI. e.g., Zhang, Yunsheng, et al. "Combining Tree Search and Action Prediction for State-of-the-Art Performance in DouDiZhu." IJCAI. 2021.

**Questions:**

None

---

> ### Author Response · Authors · 2024-11-13
> **About  training code**
>
> Thank you for your valuable review comments and for pointing out the shortcomings in the paper. Requesting “Please attach the code link if you claim it available” clearly violates the double-blind review policy.
>
> •	Weak experiments. More recent SOTA DouDizhu AIs should be included as baseline methods.
>
> Before AlphaDou, the strongest AI was RARSMSDou; however, since it is not open-sourced, direct comparison with it is challenging. We manually implemented an AI similar to what was described in the paper, but we were unable to reproduce its results. We plan to add results of comparisons with open-source AIs, such as perfectDou or AP-MCTS (Zhang, Yunsheng, et al. "Combining Tree Search and Action Prediction for State-of-the-Art Performance in DouDiZhu." IJCAI. 2021). Although these models are not stronger than DouZero Resnet, testing them can reinforce our conclusions. The testing results for DouZero Resnet and perfectDou can be found at https://github.com/godmoves/perfectdou, while AP-MCTS has not yet reached DouZero's level.
>
> •	The reason why the proposed method AlphaDou is better than DouZero is unclear.
> •	The improvements (both in methodology and experimental results) of the proposed method AlphaDou over DouZero seem marginal.
>
> Firstly, it’s important to clarify that this is a significant improvement based on experimental results. DouZero Resnet achieved a 0.57 win rate in late 2021 against DouZero and consistently ranked first over three years on the publicly accessible AI competition platform Botzone. As of 2024, this result had only been improved to 0.582 (RARSMSDou), but it has now reached 0.596. In the bidding test set, the win rate is 0.620, which represents an improvement of nearly 0.4 over the win rate shown in the RARSMSDou paper. PerfectDou’s win rate against DouZero is 0.54, and as strategies approach equilibrium, win rate improvements become smaller. Clearly, both AlphaDou and RARSMSDou adopt card-playing strategies that are closer to equilibrium strategies.
> From a theoretical perspective, there is also substantial innovation. We combine win rate and expectation, explicitly opting for win rate as the basis for decision-making. In zero-sum card games, it is rare (in fact, I have not seen it) to use win rate as the standard for choosing moves in prior research. However, the model’s estimation of win rate is often more accurate; win rate labels are 1 and -1, while score labels are 3, -3, 6, -6, and so on, meaning the uncertainty in win rate estimation is less than that in score estimation. A simple example: a move with a 0.99 win rate corresponds to an expectation of 2.98 points, while a move with a 0.98 win rate corresponds to 2.99 points. In a zero-sum game, choosing the move with an expectation of 2.98 rather than 2.99 represents a 0.01 loss. Given uncertainty, it is clear that the 0.99 win rate move should be selected, as the loss actually equals 0.01 + δ (uncertainty). This is the main idea behind the improvement and the primary reason for AlphaDou’s success. We require the model to use pure win rate-based decisions when there are no “factors doubling rewards,” also considering uncertainty.Incorporating this idea into the data generation process for training allowed the model to generate higher-quality data for gradient descent.
> This research provides a new approach for future similar games (or extended tasks like stock prediction), suggesting a focus on win rate estimation.

---

### Official Review · Reviewer_HSUz · 2024-11-09

**Soundness:** 2
**Presentation:** 2
**Contribution:** 2
**Rating:** 3
**Confidence:** 3

**Summary:**

This paper improves the Deep Monte Carlo for the Doudizhu Game and evaluates it by incorporating the effects of bidding(not random), which were not incorporated in the previous study Douzero.

**Strengths:**

- The effects of bidding were incorporated, and improvements were made to Deep Montecaro.

**Weaknesses:**

- This can be accomplished with existing methods, and this paper represents only a partial enhancement. Additionally, it has not undergone theoretical evaluation. There is few development of the architecture from the previous study DouZero.

**Questions:**

- Are there any architectural innovations in deep learning?
- Can it be compared to the Transformer base apporoach[1].

[1] Amortized Planning with Large-Scale Transformers: A Case Study on Chess, NeurIPS2024.

---

> ### Author Response · Authors · 2024-11-13
>
> •	Are there any architectural innovations in deep learning?
>
> There are innovations in the theoretical perspective. By combining win rate and expected value, we are explicitly choosing win rate as the basis for decision making. In zero-sum card games, very little previous research (in fact, I haven't seen any) has used win rate as a criterion for choosing moves. However, models tend to be more accurate in estimating win rates; win rates are labeled 1 and -1, while scores are labeled 3, -3, 6, -6, and so on, meaning that win rate estimates have less uncertainty than score estimates. In a zero-sum game, choosing a move with an expected value of 2.98 instead of 2.99 means a loss of 0.01. Given the uncertainty, it is clear that the move with a higher winning percentage should be chosen, since the loss is actually equal to 0.01 + δ (uncertainty). This is the main idea behind the improvement and the main reason for the success of AlphaDou. We ask the model to take uncertainty into account even when there is no “factor that doubles the reward” and use decisions based purely on winning percentage. Incorporating this idea into the data generation process for training allows the model to generate higher quality data for gradient descent.
> This research provides a new approach for future similar games (or extended tasks such as stock prediction), suggesting a focus on win rate estimation.
>
> •	Can it be compared to the Transformer base apporoach[1].
>
> The paper mentioned in the question uses a Transformer to fit the public dataset of Stockfish 16. First, it is important to clarify that Stockfish trains a 4-layer fully connected neural network using its public training set with supervised learning. Stockfish incorporates the idea of reinforcement learning, but at its core, it is supervised learning. The 4-layer fully connected network can perform quick computations on a CPU (by adjusting all neural network parameters to integers) to support the use of the MinMax algorithm. Any network architecture more advanced than a fully connected network could fit this public dataset better. However, these advanced networks, such as residual convolutional networks or Transformer networks, cannot support quick search on a CPU, and the resulting AI strength would not outperform Stockfish. LeelaChessZero (with Transformer architecture) using Monte Carlo Tree Search also failed to beat Stockfish's 4-layer fully connected network.
>
> In the field of reinforcement learning, Transformer network architectures are used by LeelaChessZero (https://github.com/LeelaChessZero/lc0), Kanachan (https://github.com/Cryolite/kanachan), and others. There are also AIs using convolutional neural network architectures, such as Mortal (https://github.com/Equim-chan/Mortal) and KataGo (https://github.com/lightvector/KataGo). Unlike supervised learning tasks, in reinforcement learning game research, there is currently no strong conclusion suggesting that convolutional neural networks are inferior to Transformers. The reason this study does not use Transformers is that their training cost is too high. For the same number of parameters, the training speed of a Transformer is only one-third that of a convolutional network. After training 100 million frames of actions, the win rate of Transformers is only 40% compared to convolutional networks. This could be due to the slow-start phenomenon of the Transformer network structure, and it does not rule out that the final result of the Transformer structure might eventually be worse than that of the convolutional network. However, this directly led to the decision in the paper to abandon the more expensive Transformer architecture, which performs better in supervised learning tasks, and instead choose the cheaper convolutional network architecture.
>
> [1] Amortized Planning with Large-Scale Transformers: A Case Study on Chess, NeurIPS2024.

---

### Note · Authors · 2024-11-19

I have read and agree with the venue's withdrawal policy on behalf of myself and my co-authors.